# Crowdsourcing LUCAS: Citizens Generating Reference Land Cover and Land Use Data with a Mobile App

**Juan Carlos Laso Bayas** [1,*] , **Linda See** [1] , **Hedwig Bartl** [1] , **Tobias Sturn** [1] , **Mathias Karner** [1] , **Dilek Fraisl** [1,2] , **Inian Moorthy** [1] , **Michaela Busch** [1,3] , **Marijn van der Velde** [4] and **Steffen Fritz** [1]

1   Ecosystems Services and Management Program, International Institute for Applied Systems Analysis, Schlossplatz 1, 2361 Laxenburg, Austria; see@iiasa.ac.at (L.S.); bartl@iiasa.ac.at (H.B.); sturn@iiasa.ac.at (T.S.); karner@iiasa.ac.at (M.K.); fraisl@iiasa.ac.at (D.F.); moorthy@iiasa.ac.at (I.M.); michaela.busch@gmx.at (M.B.); fritz@iiasa.ac.at (S.F)
2   Department of Economics and Social Sciences, University of Natural Resources and Life Science Vienna (BOKU), 1180 Vienna, Austria
3   Fritsch, Chiari und Partner ZT GmbH, Marxergasse 1B, 1030 Vienna, Austria
4   European Commission, Joint Research Centre (JRC), 21027 Ispra, Italy; marijn.van-der-velde@ec.europa.eu
*   Correspondence: lasobaya@iiasa.ac.at; Tel.: +43-2236-807-374

**Abstract:** There are many new land use and land cover (LULC) products emerging yet there is still a lack of in situ data for training, validation, and change detection purposes. The LUCAS (Land Use Cover Area frame Sample) survey is one of the few authoritative in situ field campaigns, which takes place every three years in European Union member countries. More recently, a study has considered whether citizen science and crowdsourcing could complement LUCAS survey data, e.g., through the FotoQuest Austria mobile app and crowdsourcing campaign. Although the data obtained from the campaign were promising when compared with authoritative LUCAS survey data, there were classes that were not well classified by the citizens. Moreover, the photographs submitted through the app were not always of sufficient quality. For these reasons, in the latest FotoQuest Go Europe 2018 campaign, several improvements were made to the app to facilitate interaction with the citizens contributing and to improve their accuracy in LULC identification. In addition to extending the locations from Austria to Europe, a change detection component (comparing land cover in 2018 to the 2015 LUCAS photographs) was added, as well as an improved LC decision tree. Furthermore, a near real-time quality assurance system was implemented to provide feedback on the distance to the target location, the LULC classes chosen and the quality of the photographs. Another modification was a monetary incentive scheme in which users received between 1 to 3 Euros for each successfully completed quest of sufficient quality. The purpose of this paper is to determine whether citizens can provide high quality in situ data on LULC through crowdsourcing that can complement LUCAS. We compared the results between the FotoQuest campaigns in 2015 and 2018 and found a significant improvement in 2018, i.e., a much higher match of LC between FotoQuest Go Europe and LUCAS. As shown by the cost comparisons with LUCAS, FotoQuest can complement LUCAS surveys by enabling continuous collection of large amounts of high quality, spatially explicit field data at a low cost.

**Keywords:** land cover; land use; citizen science; mobile apps; in-situ data collection; LUCAS

## 1. Introduction

Land cover (LC) is defined as the biophysical surface cover of the Earth, e.g., water, forest, grassland, etc. In contrast, land use (LU) is the way in which the land is used by humans or the functional aspect of the land, e.g., commercial or residential areas, grazing lands, or the types of crops grown in an area [1]. Satellite remote sensing and photo-interpretation have been used to create numerous land use and land cover (LULC) maps in the past [2]; these are used as inputs to climate, LU, and ecological models [3–5], and for calculating policy-relevant indicators, including some related to the United Nations Sustainable Development Goals (SDGs) [6].

At the European level, there are a series of LULC products that have been created as part of the Copernicus Land Monitoring Service (CLMS). The CORINE (Coordination of Information on the Environment) land cover (CLC) data set is produced every 6 years by the European Environment Agency [7]. Produced in both vector and raster format at resolutions of 100 and 250 m, CLC has been used for a diverse range of applications such as population mapping, environmental protection, and landscape planning [8–10]. The Urban Atlas contains LU data for more than 700 urban areas, i.e., cities with greater than 50 K inhabitants for 2012 and more than 300 cities with greater than 100 K inhabitants in 2006 [11]. By using the same LULC nomenclature, the Urban Atlas allows cities across the EU to be compared. Moreover, applications such as LU modeling and the calculation of various spatial metrics [12,13] are also possible. Another product of the CLMS is the High Resolution Layers (HRL) for Europe, which includes the degree of soil sealing or imperviousness, the tree cover density and forest type, grasslands, permanent water bodies and wetness, and small woody features. Some products were produced for 2012 while others have been added for the reference year 2015 [14].

Complementing these products is the LUCAS (Land Use Cover Area frame Sample) survey [15], which takes place every 3 years and is led by Eurostat. The results from this systematic survey are used for LULC change detection in European Union (EU) member countries as well as many other applications [16]. A harmonized LUCAS database with survey data and images from 2006, 2009, 2012, 2015, and 2018 has recently been published [17]. In 2015, there were 273,401 samples surveyed by 750 professional surveyors. These surveyors followed a published set of protocols for data collection at each sample point [15] with a further 67 K points photo-interpreted. In 2018, the number visited by field surveyors was 238,077 with a further 100 K points photo-interpreted [18]. The field protocol involves the surveyor travelling to the location, noting down the LU and LC using a specific nomenclature [19], and taking photographs at the point, as well as in the four cardinal directions away from the point. There are additional modules such as travelling along a transect eastward while observing any LULC changes and collecting soil samples at specific locations. More recently, the Copernicus module was added [20], which is specifically tailored for remote sensing purposes. More details can be found in the technical guides for surveyors [15,21].

It is important to note that LUCAS is the only authoritative in situ data set available for EU wide validation purposes; it contributes towards the accuracy assessment of the CLC data set and the HRLs [22–24]. Yet, in situ data on LULC could easily be collected by citizens using GPS-enabled smartphones. There are many examples of the involvement of citizens in scientific research, referred to as citizen science [25]. Observations of species, phenology, weather phenomenon, or other environmental parameters have been collected by citizens in the field [26–29]. For instance, the eBird project is one of the most successful examples of citizen science in which more than 360 million bird observations have been collected by amateur enthusiasts and made available through GBIF (Global Biodiversity Information Facility) [30].

With many citizen science projects, there are often rigorous protocols that must be adhered to, which requires training as well as commitment on the part of the citizen. There are definite tradeoffs between how complex the protocol is and keeping citizens engaged [31]. However, technology can be used to help simplify protocols. Gamification can also add an element of competition that can incentivize participation [32]. This combination of technology and competition was implemented in the summer of 2015 in a crowdsourcing campaign called FotoQuest Austria [33]. This campaign

was specifically geared towards in situ data collection of LU and LC. It adopted a simplified LUCAS protocol as the basis, and it included locations of LUCAS survey points against which the crowd could be compared, since a LUCAS survey was taking place at the same time.

The FotoQuest Austria mobile app [33] was designed to help users to fulfil the protocol as much as possible, e.g., the phone would only allow users to take photographs once the point had been reached and when the compass directions indicated the correct direction. Simple dropdown menus were also provided for choosing LU and LC. During that campaign, which ran until the end of September 2015, 2234 quests were completed at 1699 unique locations. The gamification element involved a leaderboard with users competing for prizes; these were awarded at the end of the competition and included a smartphone and tablets. The percent agreement between FotoQuest Austria and LUCAS 2015 LC and LU classes was 80% in homogeneous areas for the top-level LU and LC classes (e.g., Cropland–LC, Agriculture–LU). The LUCAS nomenclature consists of a detailed hierarchy of types that spans 3 levels of detail [33]; Table S1 in the Supplementary Information contains the complete LUCAS LC nomenclature. When the more detailed classification of LC and LU is considered, i.e., level 2 (e.g., Cereals–LC) and level 3 (e.g., Maize–LC), the agreement between the crowd and LUCAS data was much lower [33].

The 2015 campaign provided no training in LULC recognition and relied only on the knowledge of the individuals taking part in the game. The main reason was to minimize the burden on users as much as possible since each quest involved collecting data according to a protocol. There is a tradeoff between how much you can ask individuals to do and the number of users who will participate [31]. Yet, it became clear that some training was required to improve the ability of citizens to more accurately classify LULC. In 2018, the FotoQuest Go Europe campaign was launched, taking lessons learned from the 2015 campaign [33] into account. Several changes were made, e.g., improvements to the user experience (UX), a hierarchical decision tree for selecting LC, the detection of change, as well as providing near real-time feedback to the participants. The overall aim of this paper is to determine whether citizens can provide high quality in situ data by comparing the results with LUCAS data as ground truth. At the same time, this would allow us to determine whether the improvements made to the app and to the campaign in 2018 have resulted in improved quality in the in situ data collection when compared to the campaign undertaken in 2015, building upon previous work [33]. The implications of this study are relevant for the use of citizen science to complement the LUCAS survey in the future, which are discussed in the final section of the paper.

## 2. FotoQuest Go Europe

FotoQuest Austria was the first mobile app developed to examine whether citizens could classify LC and LU of points on the ground. This section outlines the next generation of this app, referred to as FotoQuest Go Europe, as well as the campaign that was undertaken.

### 2.1. The Mobile App

The FotoQuest Go Europe mobile app (Figure 1a) was developed in Unity. This is a game development environment that allows gamification features to be added easily to the app, e.g., a leaderboard as well as 3D objects. The basic idea is that users view quests on a map interface (Figure 1b). They then choose a quest and navigate towards the location on the map. The app then provides information about how far you are from the location (Figure 1c). The locations available in FotoQuest Go Europe were those employed in LUCAS 2015. Once the user comes close to the point, the app asks the user whether they can reach the point or not. If they can, it advises them to stop using the GPS and use the map to reach the point (Figure 1d).

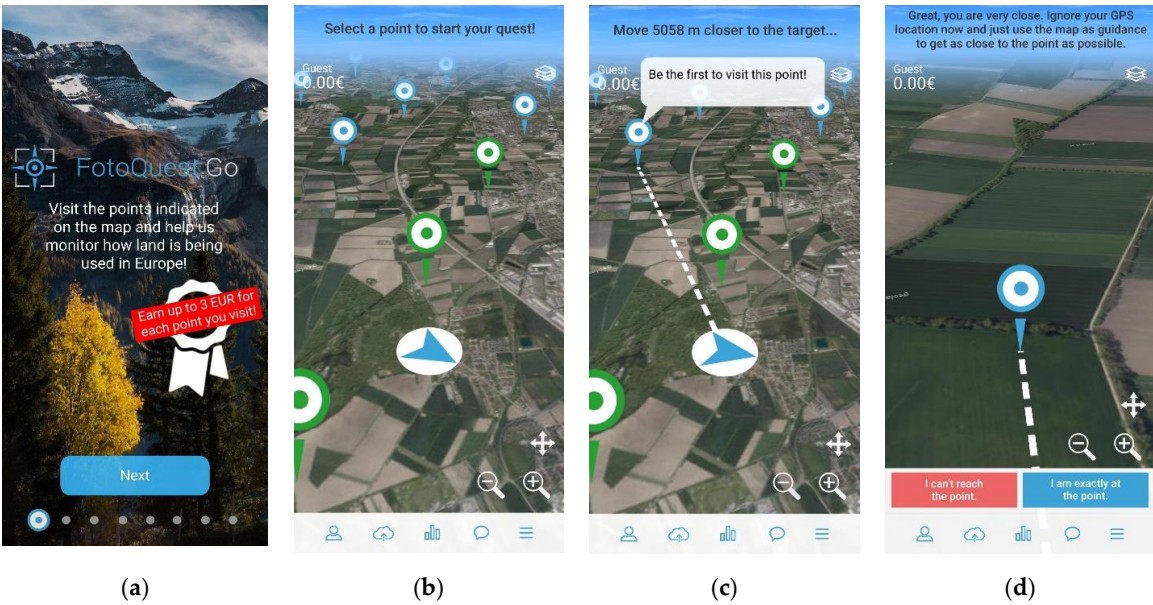

|     |     |     |     |
| --- | --- | --- | --- |
| (**a**) | (**b**) | (**c**) | (**d**) |

**Figure 1.** Screenshots from the FotoQuest Go Europe app showing (**a**) the starting screen of the app; (**b**) the map interface showing the location of quests; (**c**) a message helping the user to reach the location; and (**d**) reaching near the point.

Once the user is at the point, they can begin the quest. The quests in the FotoQuest Go Europe 2018 campaign started by displaying on-site pictures from the LUCAS 2015 campaign. Users were then asked if the LC they observed was different to the one displayed in the app. If the LC was not different, they were then requested to take photographs in the four cardinal directions away from the point and a downward looking oblique fifth picture of the location. This is the same as in the regular LUCAS protocol. The mobile app is designed to help the users take the pictures with the compass feature of the phone, only allowing photographs to be taken when the user is facing N, S, W, and E. A line drawn across the screen also helps users take photographs so that two-thirds of the photograph is land and one-third is sky as shown in Figure 2a. Additional advice about taking the photographs is also provided within the app (Figure 2b). The app also advises users not to trespass on private property, be cautious, and to observe local rules when doing a quest.

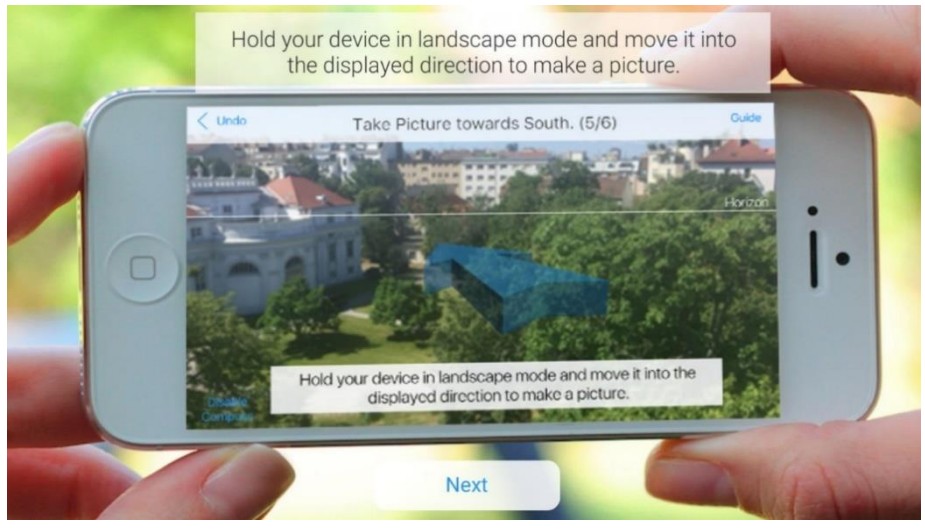

(**a**)

**Figure 2.** *Cont.*

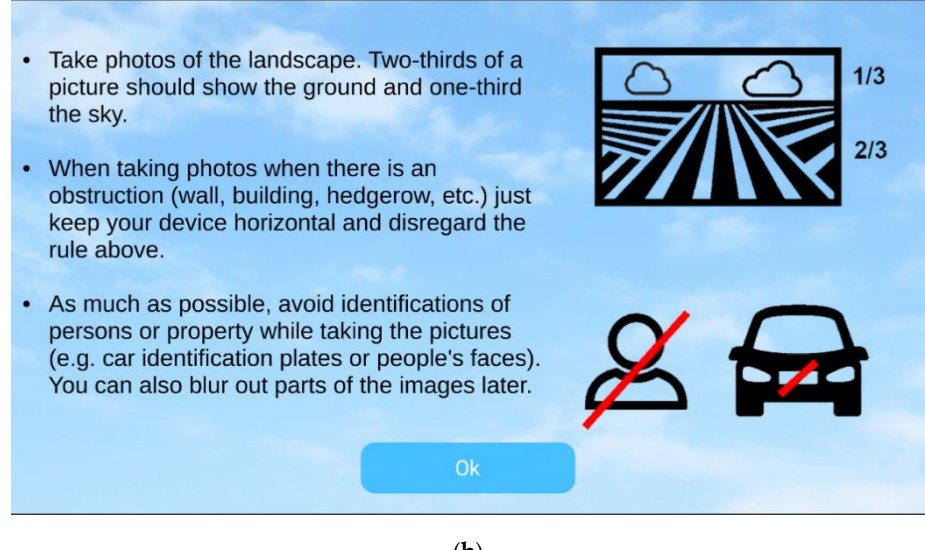

(**b**)

**Figure 2.** Screenshots from the FotoQuest Go Europe app showing (**a**) an example of a built-in feature in the mobile app that guides the user to take a photograph in one of the cardinal directions and (**b**) advice in the app about taking photographs.

If the landscape differs from the LUCAS 2015 pictures, the user is asked to classify the LC using a hierarchical decision tree represented as a series of screens with photographs. Figure 3a shows the first set of choices available in choosing the LC type. For example, if the user chooses Vegetation in Figure 3a, they will be shown a further set of LC classes to choose from as shown in Figure 3b. After LC, the user is asked to choose the LU (Figure 3c). In contrast to the 2015 FotoQuest Austria campaign, we focused on LC rather than LU. Hence, we did not employ a decision tree for LU. Rather, we asked users to select up to three LU classes from a list of 9 options. This simplified approach to LU was chosen so that participants were not overloaded with too many complex questions. After the LU was selected, the user was prompted to select how homogeneous the landscape was. The app allowed the user to select one of four available categories: <1.5, 1.5–10, 10–50, >50 m. Once these selections were made, the app requested the user to take the pictures, as explained above.

Once the pictures were taken, the quest was complete and could be submitted. Another innovation of the FotoQuest Go Europe campaign was the introduction of a near real-time quality assurance system (see Figure S1 and the detailed description in the Supplementary Information). This system was designed to provide feedback to users within 24 h of a completed quest. An expert surveyor based at the International Institute for Applied Systems Analysis (IIASA) verified each quest. This was linked to a second new feature, i.e., the payment of 1 to 3 Euros for each quest. This payment was awarded to the users if the data submitted were of sufficiently high quality. In situations where the quality was not adequate, feedback was provided to the users in a timely fashion to help them improve. Table S1 in the Supplementary Information provides examples of the types of feedback provided to the users. Table S2 summarizes the recommendations sent as feedback disaggregated by the type of feedback. The system was first piloted in a FotoQuest campaign undertaken in 2017 in Austria. It was then used in the FotoQuest Go Europe campaign undertaken in 2018.

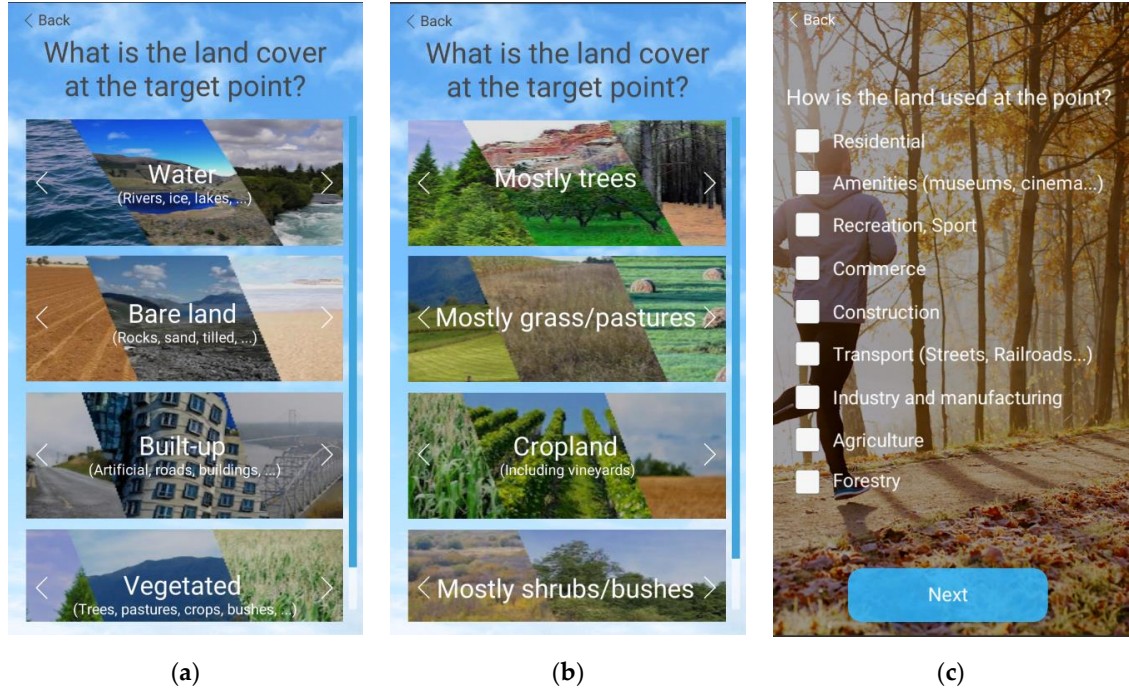

(**a**)　　　　　　　　　　　(**b**)　　　　　　　　　　　(**c**)

**Figure 3.** Screenshots from the FotoQuest Go Europe app showing (**a**,**b**) the hierarchical land cover (LC) classification and (**c**) the land use (LU) classification.

### 2.2. The FotoQuest Go Europe Campaign

The official FotoQuest Go Europe campaign ran between 8 June and 30 September 2018, although contributions were still being received afterwards. An important additional feature of the FotoQuest Go Europe 2018 when compared to the 2015 FotoQuest Austria was the wider reach of the campaign. Points were made available across Europe as LUCAS is a European-wide exercise (Figure 4).

In the 2015 campaign, prizes were awarded to the individuals that undertook the highest number of quests. In contrast, this campaign awarded 1 to 3 Euros to each successfully completed quest. Success was defined by the quality of the answers submitted, as determined through the near real-time quality assurance system (see Supplementary Information for more details). If the points were located away from a road or more inaccessible areas, they were awarded 2 to 3 Euros (depending on the difficulty). Moreover, there were weekly challenges. The first person to reach a "challenge point" and provide an answer that passed the quality assurance process would receive a one-off €30 reward. These locations were not explicitly flagged on the map. Instead, a puzzle or riddle was placed on our social media pages to help users determine the location. This was added to provide a gaming element to the campaign. The challenge locations were also sites that were relatively far or not very accessible. Additionally, a FotoQuest Go Europe point could only be visited once. This is in contrast to the 2015 campaign where the same location could be visited by different users.

Regarding LUCAS, the number of matching locations in the FotoQuest Go Europe campaign and LUCAS 2018 was 811. The campaign sample points were selected based on the 2015 LUCAS locations because the 2018 LUCAS locations were not yet available at the time of the FotoQuest Go Europe campaign. Hence, the LUCAS 2018 campaign did not include all of these locations [18].

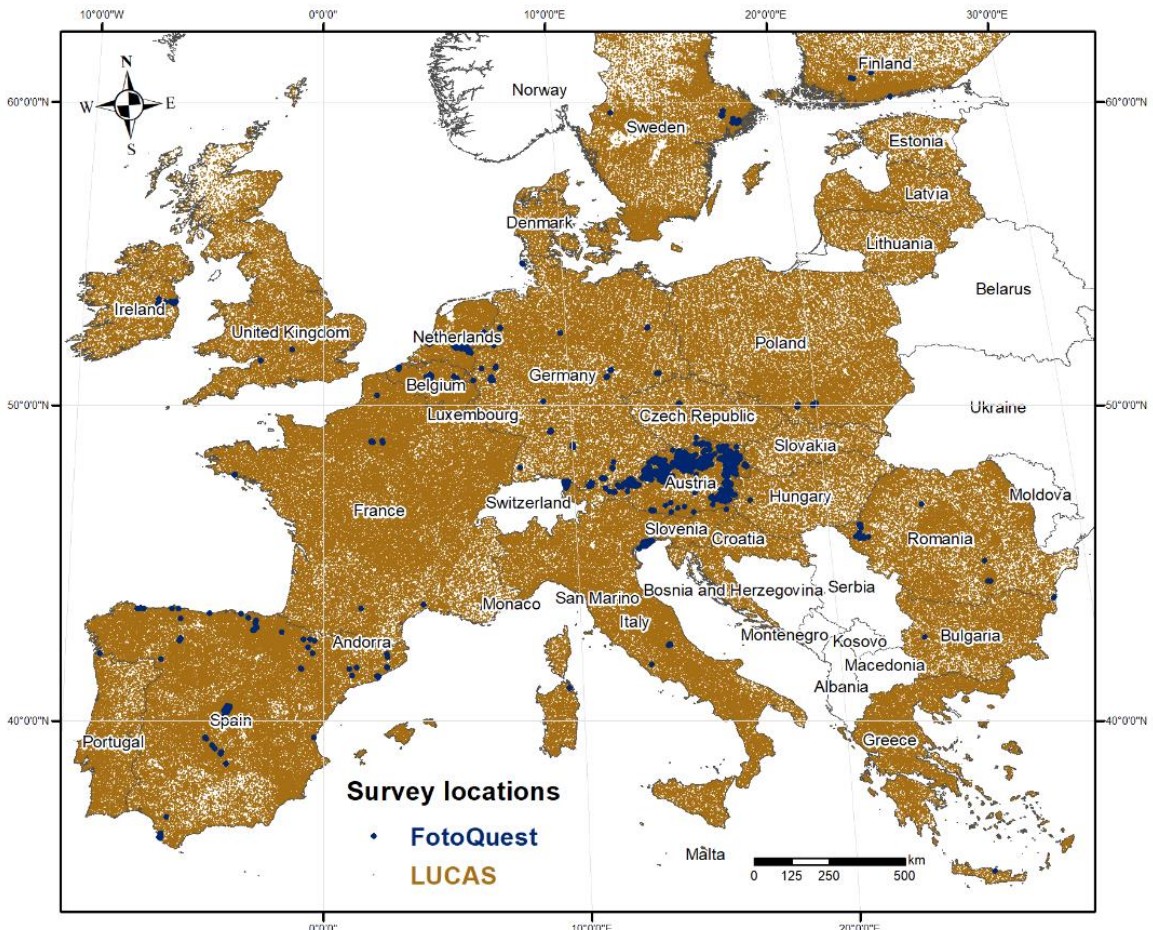

**Figure 4.** The Land Use Cover Area frame Sample (LUCAS) survey locations (in brown), showing those that were visited in the FotoQuest Go Europe campaign in dark blue (Base maps: Database of Global Administrative Areas (GADM)).

Flagged Quests

Once the campaign had started, the IIASA team received a suggestion to flag quests with potential high quality. Flagged quests are those in which the quality of the pictures was high, the proximity to the target was good, and the description of the LC was accurate. This occurred when, e.g., participants were close to the target location, pictures complied with FotoQuest standards, and the land cover description clearly matched that shown in the pictures. Therefore, from 2 July 2018 onwards (the date shortly after which the suggestion was received), quests were flagged that fulfilled these characteristics. The idea behind this characterization was to understand whether these selected quests could have significantly better agreement with LUCAS 2018.

## 3. Materials and Methods

The LC and LU reported by citizens in the FotoQuest Go Europe 2018 campaign was compared against the 2018 LUCAS data as ground truth. This was done to understand how much the data collected through FotoQuest agreed with LUCAS. Percentage agreement was calculated between FotoQuest classes and LUCAS classes. For LC, we calculated percentage agreement at each of the three different levels of LUCAS LC, e.g., level 1: Cropland, level 2: Cereals, level 3: Maize (see Table S1 in the Supplementary Information for all LUCAS LC classes and levels). For LU, percentage agreement was calculated as the match between any of the LU classes selected by FotoQuest users and the LUCAS LU classes. FotoQuest Go Europe 2018 did not include all LUCAS LU classes, just the most common

ones and only those from LU level 2. This was done purposefully so that respondents were not overwhelmed by the task and because we wanted to focus on LC instead.

For LC, we separated the percentage agreement by flagged points as well as those marked as change or no change. A confusion matrix between the 2018 LUCAS data and FotoQuest Go Europe data was tabulated to understand which classes had the most disagreement at LC level 1. Tables were then generated with percentage agreement disaggregated at different levels for the main LC classes. There is emphasis on the Cropland class because in the 2015 study, Cropland was the class with most disagreement. Therefore, we also show results related to crop type agreement. The frequency of agreement/disagreement for each year (2015 and 2018) at each level was compared using a group comparison test (chi square) where the likelihood of agreement was obtained through the Cochran–Mantel–Haenszel test.

Additionally, multivariate generalized linear mixed models were employed to understand the factors influencing agreement between data from LUCAS and FotoQuest Go Europe for 2018 for each LC level. Modeled agreement (MA) between LUCAS and FotoQuest was then defined as a binomial variable showing either agreement (MA = 1) or disagreement (MA = 0) between the LC classes from the two systems at each LC level. The models were run as binomial logistic regressions in a generalized mixed model framework with a binomial distribution. Quests selected in the models included only: (i) those where the distance between the user and the point coordinates was less than 300 m, (ii) where no feedback with a recommendation was sent (i.e., the quest was deemed correctly undertaken), and (iii) when points were skipped but still visible. Since each user provided several quests, a random effect with the user ID (UID) was included to account for lack of independence between observations coming from the same user. The models used the following as explanatory variables: Distance from the quest to the actual point coordinates (DFQ, in meters), skipping (i.e., not being able to reach the point) (SKIP, 1/0), LC homogeneity (radius: <1.5, 1.5–10, 10 –50, >50 m) (HOM, categories—1.5/10/50/100), whether the point was flagged or not (FP, 1/0), and the number of quests a given user had sent (QpU, n). Given that different users provided dissimilar amounts of data, i.e., some users visited more locations than others, the QpU variable was introduced to control for the unequal contribution of observations per user. A general representation of the model is:

$$MA = f \ (DFQ, SKIP, HOM, FP, QpU :: UID), \tag{1}$$

where the double colon separates fixed and random effects.

The Laplace estimation method was employed to fit the models in SAS (version 9.4), using Proc Glimmix. This procedure allows binomial responses but also the use of random effects. These were needed, in this case, to acknowledge repeated observations by the same user. The random effects were evaluated using statistical inference for the covariance parameters, with a significance test based on the ratio of residual likelihoods, using a "covtest glm" statement. Additionally, we compared the relative goodness of fit of the models with and without random effects. Initial correlation tests were run across all predictors to detect and avoid multicollinearity.

Finally, at the end of the campaign, two voluntary surveys (one in English and one in German) were administered to users to share their experiences and recommendations about the campaign. The results of the survey are provided in Tables S4 and S5 along with detailed explanations in the Supplementary Information.

## 4. Results

### 4.1. Results from the FotoQuest Go 2018 Campaign

In total, 140 users undertook quests covering 1612 different locations (see Figure 4). From these, a total of 71% of quests showed no change in the LC (1076 locations), 21% showed change (310 locations), and 8% were marked as not sure (118 locations).

Of the total locations visited, 637 were declared as unreachable (or skipped in the app, which was approximately 40%). In only 108 locations, it was stated that the point was not visible. Reasons given included: point on private property (50%), an obstacle was in the way (14%), or the point was inside a field with crops (10%). Other non-visibility reasons included: point on military areas (5%), point on water (4%), point in a nature area (with no access—4%), bad GPS/cell coverage (4%), point close to a highway (3%) or in heavy vegetation (3%), and various other miscellaneous reasons (3%).

Out of a total of 1021 approved quests that were submitted between 2 July 2018 and the end of the campaign, 521 were flagged quests. Of these, 74% were at locations where no LC change was reported, 20% at locations with change reported, and 6% at locations where users were not sure if change had occurred or not. Most flagged quests (56%) were submitted during the last month of the campaign (September, 293 points) with a total of 21% in August (111) and 23% in July (120).

Focusing now on LUCAS, the total number of locations in FotoQuest Go Europe that matched LUCAS 2018 was 811. At 729 of these locations, the participants were certain that a LC change or no change had occurred. At 22% of these locations, the participants of FotoQuest Go Europe reported a change in LC, i.e., differences between the LUCAS 2015 pictures and the current LC. At these same locations, LUCAS reported 31% change between LUCAS 2015 and 2018, indicating that the participants underreported the change.

The overall LC percent agreement between LUCAS 2018 and FotoQuest Go Europe for levels 1 to 3, considering all and only flagged points (marked after 2nd July), is presented in Table 1. As expected, the highest accuracies are for level 1, although accuracies are still above 74% for level 3. Moreover, flagged points do have a slightly higher agreement than all points together.

**Table 1.** LC percent agreement at different levels for all points and points flagged as potential high-quality.

| Level | Overall % (n) | Flagged Points % (n) |
|-------|---------------|----------------------|
| 3 | 74 (429) | 79 (139) |
| 2 | 82 (673) | 86 (223) |
| 1 | 90 (704) | 91 (241) |

Table 2 further disaggregates the LC percent agreement for the level 1 Cropland class and other classes. Similar patterns can be seen except for the Cropland class, where the overall accuracy is lower for levels 2 and 3 when considering all points, reflecting the difficulty in identifying crop types. However, there are notable improvements when considering flagged Cropland points. In contrast, the pattern is somewhat different for the other classes (excluding Cropland). First, the overall accuracies are much higher for levels 2 and 3. Secondly, there is little difference in the accuracies between the different levels overall as well as when considering only flagged points. This may reflect the fact that the other classes are easier to identify than the Cropland class, even when quite detailed.

**Table 2.** LC percent agreement at different levels for all points and points flagged as potential high-quality, contrasting the Cropland class and non-cropland locations.

| Level | Overall % (n) | | Flagged Points % (n) | |
|-------|---------------|----------------|----------------------|------------------|
| | In Cropland | In Other Classes | In Cropland | In Other Classes |
| 3 | 49 (174) | 91 (255) | 56 (41) | 89 (98) |
| 2 | 69 (180) | 86 (493) | 79 (43) | 88 (180) |
| 1 | 92 (200) | 89 (504) | 94 (52) | 90 (189) |

Table 3 provides the percent agreement for the three levels when considering locations that were reported as change or no change compared to 2015. In general, as the level increases (requiring a more detailed class description), the agreement decreases. The exception is in flagged points with

no change where levels 1 and 3 are similar in agreement. However, there is an observed decrease in percent agreement for level 2. The agreement of points with no change is also higher than those with change. Finally, there are small increases in the percent agreement of flagged points with no change, while the results are mixed for locations with change. In particular, for locations with change at level 3, the percent agreement of flagged locations is considerably higher than for overall locations. However, there is a slight decrease when considering level 1.

**Table 3.** LC percent agreement at different levels for all points and points flagged as potential high-quality, contrasting locations where change and no change was reported.

| Level | Overall % (n) | | Flagged Points % (n) | |
|---|---|---|---|---|
| | In Change | In No Change | In Change | In No Change |
| 3 | 55 (95) | 79 (334) | 74 (19) | 80 (120) |
| 2 | 58 (125) | 87 (548) | 63 (27) | 90 (196) |
| 1 | 78 (156) | 93 (548) | 73 (45) | 95 (196) |

Table 4 displays percent agreement levels between FotoQuest Go Europe 2018 and LUCAS 2018 for the main LC classes. These agreement levels do not represent exact agreement for individual sub-classes, e.g., Maize had higher percent agreement than Wheat, but they give a general idea of how the participants performed. The results show that both Artificial land and Woodland have high accuracies at all three levels, while accuracy decreases for Cropland and Grassland as the level increases. As these classes are amongst the most difficult to identify, this is an unsurprising result.

**Table 4.** LC percent agreement between FotoQuest Go 2018 and LUCAS 2018 for the main LC types by level. The data are sorted by the coverage of these areas in the FotoQuest Go Europe 2018 campaign in descending order.

| LC Type | Coverage in FQ Go Europe (%) | Level 1 | | Level 2 | | Level 3 | |
|---|---|---|---|---|---|---|---|
| | | n | Ag (%) | n | Ag (%) | n | Ag (%) |
| Cropland | 28 | 200 | 92 | 180 | 69 | 174 | 49 |
| Artificial land | 23 | 165 | 94 | 164 | 90 | 153 | 93 |
| Grassland | 22 | 154 | 81 | 151 | 77 | 0 | - |
| Woodland | 22 | 153 | 97 | 149 | 95 | 90 | 90 |
| Others | 5 | 35 | 43–80 | 32 | 43–80 | 13 | 75.89 * |

\* Wetlands and Water, respectively. No observations for the Shrubland or Bareland classes.

The LC confusion matrix (Table 5) shows that the most common confusion was between Grassland and Cropland, where Cropland in LUCAS 2018 is, at times, confused with Grassland (Table 5). Compared to the 2015 FotoQuest Austria result (Table 2, [33]), the user accuracy increased in the 2018 campaign in all classes, especially for Artificial land class (94% compared to 54% in 2015) and Shrubland (71% compared to 14% in 2015).

Overall percent agreement between the FotoQuest Go 2018 results and LUCAS 2018 for the main crop types is shown in Table 6. The highest percent agreement is for Maize, which is a relatively easy crop to identify followed by Soya. Common wheat had less than 50% agreement, but this may be due to confusion with other cereals that look similar, e.g. the Barley class.

In terms of LU, the overall percent agreement was 65%. Table 7 shows the agreement by the main LU types. Some LU types have high percent agreement, in particular, Agriculture, Forestry, and Construction, while Commerce had just over 50% agreement. Interestingly, Road transport had 0 percent agreement. This may be due to the fact that participants were on a road, but they may have looked beyond the point to capture the general LU in the surrounding area (indicated by feedback provided to users in the near real-time system—see SI). Alternatively, they did not reach the point with sufficient accuracy since the coverage of this LU type is small in area compared to other LU types.

**Table 5.** Confusion matrix showing level 1 LC classifications from FotoQuest Go Europe and LUCAS 2018.

| | | Artificial | Cropland | Woodland | Shrubland | Grassland | Bareland | Water | Wetland | User Acc. (%) |
|---|---|---|---|---|---|---|---|---|---|---|
| **FotoQuest Go Europe** | Artificial | 157 | 1 | 6 | 0 | 3 | 0 | 0 | 0 | 94 |
| | Cropland | 3 | 184 | 1 | 0 | 8 | 6 | 0 | 0 | 91 |
| | Woodland | 0 | 0 | 149 | 2 | 1 | 2 | 0 | 0 | 97 |
| | Shrubland | 1 | 0 | 1 | 10 | 2 | 0 | 0 | 0 | 71 |
| | Grassland | 8 | 14 | 9 | 2 | 121 | 0 | 0 | 0 | 79 |
| | Bare land | 0 | 4 | 0 | 0 | 0 | 3 | 0 | 0 | 43 |
| | Water | 0 | 0 | 0 | 0 | 1 | 0 | 8 | 0 | 89 |
| | Wetland | 1 | 0 | 0 | 0 | 0 | 0 | 0 | 3 | 75 |
| | Prod. Acc. (%) | 92 | 92 | 86 | 94 | 82 | 0 | 100 | 100 | |

**Table 6.** Percent agreement in the main LC class Cropland by crop type. The data are sorted according to percentage of agreement in descending order.

| Crop Type at Level 3, Cropland | n | Agreement (%) |
|---|---|---|
| Maize | 49 | 80 |
| Soya | 11 | 64 |
| Common wheat | 32 | 47 |
| Sugar beet | 7 | 43 |
| Temporary grasslands | 13 | 38 |
| Barley | 10 | 20 |
| Dry pulses | 9 | 0 |
| Durum wheat | 5 | 0 |
| Others | 1–5 | 0–100 * |

* One observation on "Other root crops": 100% agreement.

**Table 7.** Percent agreement by the main LU types. The data are sorted by the coverage of these areas in the FotoQuest Go Europe 2018 campaign in descending order.

| LU Type | Coverage in FotoQuest Go Europe (%) | N | Ag (%) |
|---|---|---|---|
| Agriculture | 40 | 309 | 93 |
| Forestry | 18 | 135 | 89 |
| Road transport | 12 | 91 | 0 |
| Construction | 12 | 78 | 87 |
| Semi-natural and natural areas not in use | 5 | 33 | 0 |
| Amenities; museums; leisure | 4 | 33 | 6 |
| Commerce | 2 | 15 | 53 |
| Others | 6 | 48 | 0 |

*4.2. Percent Agreement between the FotoQuest 2015 and 2018 Campaigns*

When comparing the 2015 and 2018 campaigns, the results of the Cochran–Mantel–Haenszel test showed significant differences between the campaigns for each LC level (1–3) (Figure 5), with higher levels of percent agreement found in the 2018 campaign. In this campaign, citizens were 2.9 times more likely to agree with LUCAS 2018 for the LC level 3 ($p < 0.001$, n = 696), 3.5 times for the LC level 2 ($p < 0.001$, n = 955), and 3.1 times for the LC level 1 ($p < 0.001$, n = 1006). Hence, there were considerable improvements in the 2018 FotoQuest campaign compared to 2015. Note that the test results and agreement ratios are valid only for comparisons at each LC level but not between levels (and hence the different letter notation in Figure 5—for LC level 1: a/b; for LC level 2: A/B; for LC level 3 aa/bb).

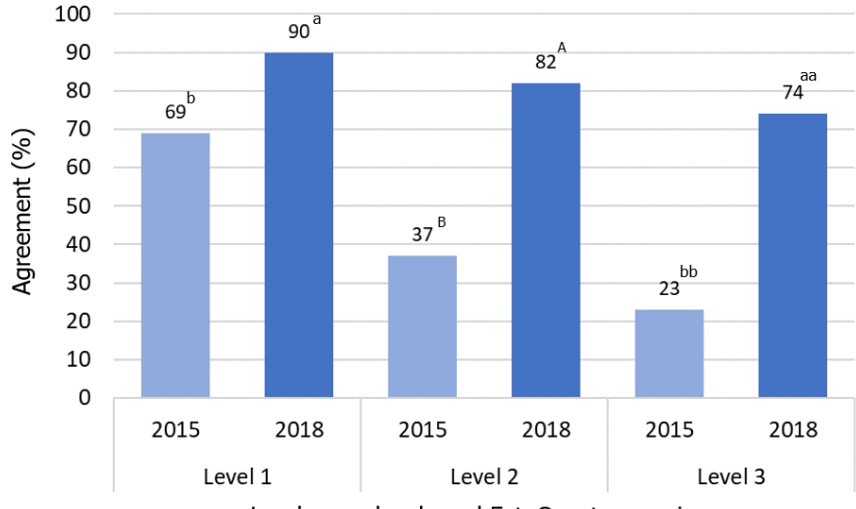

**Figure 5.** Comparison of percent agreement between LUCAS and FotoQuest for the 2015 and 2018 campaigns. Different letters show significantly different likelihoods of agreement per level (chi square test).

*4.3. Multivariate Analysis of LC Modeled Agreement (MA)*

At LC level 1, the binomial linear mixed models showed that there were no significant factors influencing the modeled agreement between LUCAS 2018 and FotoQuest Go Europe ($p > 0.05$, n = 583). However, in the model run at LC level 2, the land cover homogeneity (HOM), significantly increased the odds of agreement between LUCAS 2018 and FotoQuest Go Europe ($p = 0.0244$, n = 530). A quest with a selected homogeneity greater than 50 m increased the odds of agreeing with LUCAS 2018 by a factor of 2.3 to 2.5 times when compared to one with a selected homogeneity of between 1.5 and 10 m ($p = 0.02$) or less than 1.5 m ($p = 0.01$), respectively. Other comparisons were not significantly different ($p > 0.05$), and all other variables were not significant ($p > 0.05$).

Finally, at LC level 3, distance to the point (DFQ) was the only significant predictor of modeled agreement between LUCAS 2018 and FotoQuest Go Europe ($p = 0.023$, n = 261). This means that for every 10 m closer to the point, the odds of such a quest agreeing with LUCAS increased by a factor of roughly 1.1. All other variables were not significant ($p > 0.05$).

**5. Discussion**

One of the most important conclusions from the overall FotoQuest experience is that the increased user-guidance provided in the FotoQuest Go Europe app (in 2018) compared to the FotoQuest Austria app (in 2015) has made a considerable difference to the accuracy of LULC identification by citizens. The significant differences found between the percent agreement with LUCAS achieved across the two campaigns (2015 and 2018) are a clear indication of the potential of citizen science for in situ monitoring of LULC. These findings are in line with other citizen science projects that have either provided inputs to authoritative databases or are a proof of concept to illustrate how data from citizen science could be ingested within official systems. For example, there are many citizen science projects that provide species data to the Global Biodiversity Information Facility (GBIF). Chandler et al. [34] estimated that more than 40% of GBIF data, which are used for scientific studies and international reporting on biodiversity indicators, come from citizen science projects. The UN SDG indicators use data from citizen science to monitor progress on the SDGs [35]. On the LULC side, Liu et al. have shown how data collected through a web-based and mobile app called Paysages can be integrated with data fusion approaches into the official LULC map of the French National Mapping Agency [36]. Hence, there is considerable interest in using data from citizen science in official data sets.

Various landscape related topics (e.g., landscape features, carbon storage) could benefit from the high-quality in situ data that FotoQuest could generate. It could also become a reliable technology for increasing the amount of training data for the production of European land cover monitoring products such as CLC+ [37], which is the second generation of CLC and will contribute to LULC monitoring in the EU for decades to come, the high resolution layers, and the local components (e.g., the Urban Atlas) to complement the LUCAS survey. Complementing can be through increased temporal resolution, since LUCAS occurs every three years, increased spatial resolution, i.e., providing a denser sample, or by including locations where LUCAS is currently too expensive, e.g. above 1000 m. These advantages have been flagged as benefits of data from citizen science [38].

Although a cost comparison between LUCAS and FotoQuest is not that straightforward due to the different nature and approach of the two systems, it is illustrative to see the cost advantages of crowdsourcing. For example, the FotoQuest Go Europe 2018 field data collection and quality control costs (including the time of the expert surveyor providing near real-time feedback) are around € 2.60 per point, whereas based on the 2015 LUCAS field work tender costs (lots 1 to 5) [39] and the amount of points collected (339,697) [40], LUCAS costs are around €32.40 per point. However, it should be noted that at the current stage, FotoQuest is not intended to replace a professional survey such as LUCAS, which also collects additional variables, but its temporal and spatial complementarity to official data, reliability, accessibility, and low cost make it a worthwhile additional investment for enhancing in situ data on LULC.

Additionally, FotoQuest can connect citizens to Earth Observation, in general, but also more specifically to the Copernicus program through raising awareness and demonstrating to citizens how their data is used for map production. The user survey (see Supplementary Information) confirmed the findings that the app's user-friendliness as well as the additional pictures to aid LC identification and the guiding links were useful. Moreover, the request by many users to implement additional features such as in-app navigation and offline access to maps indicates that there is a potential for continuing to improve and use the FotoQuest technology for in situ LULC monitoring in the future.

From the results of the binomial linear mixed models, the significant effect of distance to the point (DFQ) on modeled agreement with LUCAS for level 3 LC confirms the need to improve the navigation features. The significant effect of the homogeneity of the point (HOM) for level 2 LC, where users had better percent agreement in homogeneous points compared to less homogeneous points, indicates that we could further improve the in-app support for users to identify different LC types. This is especially true for cropland, which had the lowest percent agreement amongst all classes. Perhaps an AI-enhanced system, i.e., a system that provides automated crop identification to the user in real-time using techniques from computer vision, could help to improve identification of this difficult LC type. There is a considerable amount of research being undertaken in this area, e.g., [41], which could be integrated into FotoQuest in the future.

In terms of LU, one important difference between the 2015 FotoQuest campaign and the FotoQuest Go Europe 2018 campaign was that the LU choice was simplified, i.e., users no longer selected different levels of LU. LU options were shown as a list of 9 choices (see Figure 3c), where users could select up to three different options. This was done on purpose so that we could focus on LC and not overload the user with a long decision process per quest. From a UX point of view, the process had to be simple and not take too long. Moreover, while taking pictures and identifying the LC, users were simultaneously considering the LU, which could then be filled in at the end of the quest. Finally, in-app improvements to identify the Road transport LU type (currently poorly identified) could be added in the future. In fact, the whole purpose of the app can be directed towards LU if required. As the 2018 FotoQuest Go Europe campaign has shown, an increased UX promotes high quality results for LC.

The improvements made to the app, including the user-friendly interface, guidance for taking the pictures, a visually-enhanced decision support system for identifying the LC, and the near real-time feedback system, resulted in users getting closer to the target point and obtaining higher quality pictures although there were still a number of users that were quite far from the location or submitted

poor quality photographs. Additional AI systems, which can provide alerts and suggestions for how to improve the quality of the picture taken, could be easily implemented in the app. The quality assurance service being developed in the LandSense project already addresses some of these issues [42].

Although further analyses are being done to understand potential effects from the feedback system in terms of improved agreement with LUCAS, the main benefit of such a system is that it facilitates a strong interaction between the team at IIASA and the users. The results of the user survey administered at the end of the campaign (see Supplementary Information) indicated that people reacted better when they felt involved, with some of the users criticizing the "default" messages, i.e., they would have liked even more personalized guidance. This finding is clearly aligned with the fact that good feedback and communication have been previously recognized to be of high importance in citizen science projects [43].

It is worth mentioning that users thought that push notifications of challenge points or other better methods of advertisement could have driven up the number of users and increased the gaming effect. This campaign (2018) as well as the 2015 campaign had some media coverage via television and advertisement through the Geo-Wiki newsletter, a regular publication from the Center for Earth Observation and Citizen Science at IIASA, as well as limited promotion via social media. Citizen engagement and retention are recurring issues for citizen science projects in general [44] and FotoQuest in particular, and one we believe would benefit from further support from higher level organizations, e.g., the European Commission or the European Environment Agency, given the potential of the technology to achieve very high quality results while involving citizens across Europe.

It should be noted that the study is limited to percent and modeled agreement as defined above. Methodologically speaking, the performance of FotoQuest could also have been analyzed in a different way, e.g., proximity to the target, the number of citizens visiting hard to access points or locations that were only photo-interpreted in LUCAS, etc. We are currently looking at alternative measures of system performance and potential improvements for future campaigns. Furthermore, a limitation of the employed analysis (i.e., generalized linear mixed models) is that with large sets of data, the computations take quite some time and the models may not converge, which could be a problem in the future when a larger amount of data is collected by citizens.

## 6. Conclusions and Outlook

FotoQuest is a LULC monitoring activity complementary to LUCAS. Following the detailed assessment provided here, we have shown that high quality in situ data can be gathered, and that citizens may more easily monitor certain LC types. FotoQuest type activities can be carried out each year, or in those years that LUCAS is not planned. Furthermore, FotoQuest campaigns may be tailored to specific validation tasks, e.g., related to specific HRLs. The FotoQuest Go Europe data presented here is open and freely available, thus contributing—as LUCAS—to much needed common sets of reference data across the EU to benchmark a variety of commercial LULC products.

Additionally, since one of the aims of FotoQuest Go Europe was to engage citizens in tracking LULC change over time, it can contribute to the monitoring of the SDGs, which is considered to be a key priority at the EU level [45]. From a recent detailed mapping of citizen science projects to SDG indicators [35], FotoQuest was highlighted as a citizen science project that could potentially contribute to six SDG indicators, i.e., 2.4.1 Proportion of agricultural area under productive and sustainable agriculture, 6.6.1 Change in the extent of water-related ecosystems over time, 15.1.1 Forest area as a proportion of total land area, 15.2.1 Progress towards sustainable forest management, 11.3.1 Ratio of land consumption rate to population growth rate, and 15.4.2 Mountain Green Cover Index. Hence, FotoQuest technology could be used for a variety of EU monitoring activities.

The data collected in FotoQuest campaigns includes the description and classification of LC and LU at various locations but FotoQuest users also produced a vast library of in situ photographs. Analysis of the quality and usability of these photographs and their potential use as reference data is already being undertaken at IIASA, especially since the pictures taken combined with the information provided

by reliable users can be of massive importance. We are also investigating whether payments per point or to top validators (as done in the 2015 campaign—[33]) produce different effects, e.g., change the type of location being visited (far/close to roads). Moreover, we can obtain multiple observations at the same location (demonstrated in the 2015 campaign although not used in the 2018 one), which can be used to ensure higher quality via consensus methods or other ways of combining observations. The benefits of a funded, ongoing FotoQuest campaign driven by citizens, who, amongst other reasons, enjoy being outdoors and contributing to science, could be harnessed as a steady and reliable provision of reference data to ground truth the ever-increasing amount of remote sensing big data now available and the new generation of LULC products that they will spawn.

**Supplementary Materials:** The following are available online at http://www.mdpi.com/2073-445X/9/11/446/s1, Supplementary materials contain one figure, 5 tables, a section further describing the near real-time quality assurance system, and a section detailing the results from the user surveys. Figure S1: The FotoQuest quality branch in the Geo-Wiki platform showing the actual LUCAS point (Target), the location where the LUCAS surveyor did their survey in 2015 (Lucas) and the location of the FotoQuest Go Europe user (FotoQuest). Pictures from the LUCAS 2015 survey are displayed along with those submitted by the user. A messaging system allows the FotoQuest team to deliver feedback directly to the user in almost real-time as shown in the 'Message sent' box, Table S1: LUCAS LC classes for levels 1, 2, and 3, Table S2: Types of feedback provided to the users from the near real-time quality assurance system, Table S3: Recommendations sent as feedback disaggregated by type of feedback, Table S4: Results from the survey regarding which features of the FotoQuest Go Europe 2018 campaign were liked by the users. An intense green color means a higher number of respondents, whereas pale blue colors represent the lowest number of respondents, Table S5: User-recommended improvements for future FotoQuest Go Europe campaigns. An intense green color means a higher number of respondents, whereas pale blue colors represent the lowest number of respondents.

**Author Contributions:** Conceptualization, J.C.L.B., L.S., D.F., I.M. and S.F.; Data curation, J.C.L.B., T.S., M.K. and I.M.; Formal analysis, J.C.L.B. and H.B.; Funding acquisition, L.S., I.M. and S.F.; Investigation, J.C.L.B. and H.B.; Methodology, J.C.L.B., L.S., T.S. and S.F.; Project administration, L.S., D.F. and S.F.; Resources, L.S., I.M. and S.F.; Software, T.S. and M.K.; Supervision, J.C.L.B., L.S. and S.F.; Validation, M.K. and M.B.; Visualization, J.C.L.B., L.S. and T.S.; Writing—original draft, J.C.L.B. and L.S.; Writing—review & editing, J.C.L.B., L.S., H.B., T.S., M.K., D.F., I.M., M.B., M.v.d.V. and S.F. All authors have read and agreed to the published version of the manuscript.

**Funding:** This research was funded by the EU FP7 ERC CrowdLand project, grant number 617754, and the EU Horizon2020 LandSense project, grant number 689812. The APC was funded by the EU Horizon2020 LandSense project, grant number 689812.

**Acknowledgments:** We would like to thank all the participants who took part in the FotoQuest Go Europe campaign as well as those in previous campaigns. Without your contributions, this research would not be possible.

**Conflicts of Interest:** The authors declare no conflict of interest. The funders had no role in the design of the study; in the collection, analyses, or interpretation of data; in the writing of the manuscript, or in the decision to publish the results.

## Abbreviations

| | |
|---|---|
| CLC | CORINE Land Cover |
| CLMS | Copernicus Land Monitoring Service |
| CORINE | Coordination of Information on the Environment |
| EU | European Union |
| GPS | Global Positioning System |
| HRL | High Resolution Layer |
| IIASA | International Institute for Applied Systems Analysis |
| LC | Land Cover |
| LU | Land Use |
| LUCAS | Land Use Cover Area frame Survey |
| LULC | Land Use Land Cover |
| SDG | Sustainable Development Goal |
| UX | User Experience |

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
