# Peer review of "Crowdsourcing LUCAS: Citizens Generating Reference Land Cover and Land Use Data with a Mobile App"

_land, doi:10.3390/land9110446_

Round 1

Reviewer 1 Report

Overall, the manuscript presents a novel and timely contribution to both land cover land use mapping data and crowdsourcing efforts in an organized and well documented approach. The authors leverage existing in-situ data to provide the often limited ground truth for evaluating citizen science studies, as well as address some of the major challenges with citizen science efforts aimed at expanding our ground truth data, specifically data quality and participant engagement.

One of the main strengths of the paper -- in addition to the well thought out campaign and effort to evaluate classification accuracy drivers -- is how the various components of the research support and inform others; from the app interface designed to increase participation and data quality to the near real-time feedback and a post-participation survey. Connecting the model results and survey responses to analysis outcomes and proposed enhancements/next steps is an asset to the discussion. 

A few additional details in the methods and results section could improve those connections and promote similar adoptions/replications where data are available, including:

  • End of section 1: A brief mention here about how crowdsourced and LUCAS locations are determined (i.e., crowdsourced always aligned with surveyor points from 2015, LUCAS determined with 2km x 2km grid) would be helpful to connect to section 2. Also would suggest mention of how private property access is handled before reference on page 7 as some of the example photos highlight the potential point in an agricultural field. 
  • Section 2.2:
    • Recommend adding a sentence about who provided feedback or reference that details are in section 2.3. For example, FotoQuest team was mentioned in the caption of Figure 5, but it would be helpful to include in the body of the text as well.
    • Also, what was the estimated time commitment from team members (and was it one reviewer per photo) to provide feedback? These details would assist readers with gauging feasibility for similar implementations. Was part of this process automated based on distance thresholds? 
    • With the inclusion of the high quality flag, additional details on how data were classified (i.e., photos high quality because proportions of sky versus land, completeness, distance to target threshold) would aid in future replication as well as understanding potential reviewer agreement and influence on model results.  
  • pg. 5 Ln 164 -- can a user select multiple land uses? The discussion mentions up to 3, but would be helpful to know earlier to frame the results. If a user is unsure, was there guidance in the training?
  • pg. 7 line 211 -- recommend spelling out International Institute for Applied Systems Analysis for the first IIASA reference
  • pg. 9 -- was there a distance threshold for sending a recommendation about moving closer to the surveyor point? This could be included as a footnote to table 2 or in the body of the text.  
  • pg. 9 line 260 -- For the 46 users not receiving feedback, was it due to always submitting complete and successful responses?
  • pg 10 line 292 -- the equation appears to be missing the closing parentheses
  • Discussion
    • pg 16 -- Several use cases are given for including FotoQuest reference data. Can the authors comment on the considerations for integrating FotoQuest crowdsourced data with other data sources (e.g., agency, research, etc)? Should there be quality tags like those in other citizen science platforms like iNaturalist?
  • General questions:
    • Did users report challenges with GPS coverage with mobile devices? This may contribute to the difference between surveyor location and observer location and worth noting potential positioning accuracy differences between devices used by surveyors and those by participants. 
    • For tables 4 and 5, are results presented for samples taken after 2 July when the high quality flag was included? Recommend including a footnote to tables that include data captured between 8 June and 2 July (or conversely, that only include data with the high quality flag). 

Author Response

We would like to thank the reviewer for their very useful comments, which have helped to improve the paper. The reviewers’ comments are in bold followed by our response.

Overall, the manuscript presents a novel and timely contribution to both land cover land use mapping data and crowdsourcing efforts in an organized and well documented approach. The authors leverage existing in-situ data to provide the often limited ground truth for evaluating citizen science studies, as well as address some of the major challenges with citizen science efforts aimed at expanding our ground truth data, specifically data quality and participant engagement.

One of the main strengths of the paper -- in addition to the well thought out campaign and effort to evaluate classification accuracy drivers -- is how the various components of the research support and inform others; from the app interface designed to increase participation and data quality to the near real-time feedback and a post-participation survey. Connecting the model results and survey responses to analysis outcomes and proposed enhancements/next steps is an asset to the discussion. 

Response: We thank the reviewer for their comments and summary.

A few additional details in the methods and results section could improve those connections and promote similar adoptions/replications where data are available, including:

End of section 1: A brief mention here about how crowdsourced and LUCAS locations are determined (i.e., crowdsourced always aligned with surveyor points from 2015, LUCAS determined with 2km x 2km grid) would be helpful to connect to section 2. Also would suggest mention of how private property access is handled before reference on page 7 as some of the example photos highlight the potential point in an agricultural field. 

Response:

We have added some text on how the two systems (LUCAS and FotoQuest) share locations and class definitions in the last paragraph of section 1. Additionally, since the app warns users not to enter private property, we added a sentence before Figure 1.

Section 2.2:

Recommend adding a sentence about who provided feedback or reference that details are in section 2.3. For example, FotoQuest team was mentioned in the caption of Figure 5, but it would be helpful to include in the body of the text as well.

Response:

As suggested by reviewer #3, we have tried to focus the paper a bit more, emphasizing the comparison with LUCAS. The feedback system details are shown in the supplementary information and we have clarified who provided the feedback (one professional surveyor based at IIASA and also an author of the paper) in the last paragraph of section 2.1, after Figure 3.

Also, what was the estimated time commitment from team members (and was it one reviewer per photo) to provide feedback? These details would assist readers with gauging feasibility for similar implementations. Was part of this process automated based on distance thresholds? 

Response:

As mentioned in the previous answer, the professional surveyor provided feedback via a temporary full-time contract for the duration of the campaign. This is included in the cost calculation mentioned in the last section of our manuscript.

With the inclusion of the high quality flag, additional details on how data were classified (i.e., photos high quality because proportions of sky versus land, completeness, distance to target threshold) would aid in future replication as well as understanding potential reviewer agreement and influence on model results.  

Response:

As suggested by reviewer #3, the high-quality points are now called “flagged quests”. At the end of section 2, we now include a further description of how these locations were flagged, “e.g., participants were close to the target location, pictures complied with FotoQuest standards, and the land cover description clearly matched that shown in the pictures”.

pg. 5 Ln 164 -- can a user select multiple land uses? The discussion mentions up to 3, but would be helpful to know earlier to frame the results. If a user is unsure, was there guidance in the training?

Response:

We have clarified this in the paragraph after Figure 2: “... participants were only asked to select up to three LU classes from a list of 9 potential land uses.”

pg. 7 line 211 -- recommend spelling out International Institute for Applied Systems Analysis for the first IIASA reference

Response: We have corrected this in the last paragraph of section 2.1.

pg. 9 -- was there a distance threshold for sending a recommendation about moving closer to the surveyor point? This could be included as a footnote to table 2 or in the body of the text.

Response:

No threshold was set. The recommendation was sent if it was apparent that the participant could get closer to the point but did not. This was based on visual inspection of the location coordinates against the quest coordinates and the submitted pictures.

pg. 9 line 260 -- For the 46 users not receiving feedback, was it due to always submitting complete and successful responses?

Response:

After the new Table S1 (previously Table 1), we state that a total of 94 users received recommendations (as defined in Table S1). Nevertheless, all users received feedback. However, some received only neutral or motivational feedback but with no specific recommendations.

pg 10 line 292 -- the equation appears to be missing the closing parentheses

Response: Thanks for this observation. The parenthesis is now added.

Discussion
pg 16 -- Several use cases are given for including FotoQuest reference data. Can the authors comment on the considerations for integrating FotoQuest crowdsourced data with other data sources (e.g., agency, research, etc)? Should there be quality tags like those in other citizen science platforms like iNaturalist?

Response:

At IIASA we have experimented with several different ways of crowdsourcing land cover and land use data, and we would definitively argue for including quality tags associated with the data collected. In the case of FotoQuest, if it were to be used on its own and integrated with other data sources, a quality tag could be assigned based on agreement between users who visit the same location multiple times. Another quality tag could be based on the past performance of the participants, e.g., greater confidence could be assigned to those observations from individuals with a higher past performance. We could also implement some of the best practices from iNaturalist, e.g., including peer and expert review of information submitted.

General questions:

Did users report challenges with GPS coverage with mobile devices? This may contribute to the difference between surveyor location and observer location and worth noting potential positioning accuracy differences between devices used by surveyors and those by participants. 

Response:

No, in general, users did not report challenges with GPS coverage to us. However, in the third paragraph after Figure 3, we list the reasons why some locations were skipped, where out of the total locations skipped, only 4% were reported as bad GPS/cell coverage. In other examples, the distance to the point was influenced not by the accuracy of the GPS but by how close a user could get to a location, e.g., a point located in the middle of a private field where access from the nearest road was the closest that the user could get to that location. We did retrieve the information recorded by the phone on the GPS accuracy and the median value across all of our FotoQuest campaigns is 5m.

For tables 4 and 5, are results presented for samples taken after 2 July when the high quality flag was included? Recommend including a footnote to tables that include data captured between 8 June and 2 July (or conversely, that only include data with the high quality flag). 

Response:

On the “flagged quests” section at the end of section 2, we describe the dates where these points come from. In the paragraph before the previous Table 4 (now Table 2), we mention that we compare all and then only flagged points, and we have added “(marked after 2nd July)” to emphasize this. The paragraph before the previous Table 5 (now Table 3) mentions that it further disaggregates the information in Table 2, so we think that this clarification addresses your comment.

Reviewer 2 Report

This research reported the experience and results from FotoQuest Go Europe 2018, which aims to obtain LULC information via crowdsourcing. With this topic, it is of interest to this journal. However, there are still several issues demanding further discussion and clarification. Before accepting it, I suggest that authors consider the following for revision:

Major

1. Line 100: The 3-level hierarchy structure of the LC is not clear, which can be explained more in details with proper examples/tables.

2. Line 217: A near-realtime quality assurance system has been introduced, however, it is not clear, are they operated by experts or also via crowdsourcing? If by experts, how much work efforts are needed to fulfil the goal reply within 24 hours? If via crowdsourcing, how good the users can perform such a three-level LULC classification task based on images?

3. Line 292: please check whether the formula (1) is correctly presented with only one left bracket. Since a formula presented with the double colon is also not often to be seen, is there any other research also used such form to present random/fixed effect?

4. From formula 1, are the categorical variables (e.g. SKIP, HOM) and continuous variables (e.g. DFQ, QpU) considered differently for the evaluation? If yes, how? If not, what is the reason they do not need to be separately considered?

5. Line 324: the reason why cropland has been paid special attention than others is not clear at the beginning, which should be explained before presenting the results.

6. Line 360: what happened to the rest of the other classes? Please also provide the confusion matrix from 2015 in order to present a complete performance comparison for all classes.

7. Line 404 & 407: the results from other variables are mentioned in a less detailed way. A table/figure would be helpful to have an overview of the influence of all the variables. Further discussion of the reasons is needed to provide more insights from the authors.

8. Line 458: "Corine +" was mentioned without any further explanation, though "CORINE" was mentioned in line 49. What are the difference?

9. Line 274: Is Cochran-Mantel-Haenszel test the only choice to conduct a group comparison test? If not, the reason why this test was more suitable than others need to be clarified.

10. Figure 6: the notions a/b/A/B/aa/bb needs to be explained in the text. It is also not clear the relationship between the mentioned ratios (2.9, 3.5, 3.1) and the numbers in this figure. More explanations are needed for the notions and numbers in this figure.

Minors

11. Since there are many abbreviations appeared in the manuscript, a list of abbreviations (as in the template) will be very helpful for readers to track the meaning of these abbreviations.

12. There are words with typos and inconsistent capital/lowercase letters, please have a detailed check across the whole text. E.g. Line 292 OpU/QpU, Euros/euros are not written consistently.

13. "Section 4.1" appeared twice at Line 385 and Line 306

14. Line 343: Bottom line is missing for Table 5

Author Response

We would like to thank the reviewer for their very useful comments, which have helped to improve the paper. The reviewers’ comments are in bold followed by our response.

This research reported the experience and results from FotoQuest Go Europe 2018, which aims to obtain LULC information via crowdsourcing. With this topic, it is of interest to this journal.

Response: Thank you for the positive comment.

However, there are still several issues demanding further discussion and clarification. Before accepting it, I suggest that authors consider the following for revision:

Major

  1. Line 100: The 3-level hierarchy structure of the LC is not clear, which can be explained more in details with proper examples/tables.

Response:

In the text, we have introduced a simple explanation with examples (highlighted here for clarity):

 “When compared with the authoritative data from LUCAS, the results showed agreement of 80% in homogeneous areas for the top-level LU and LC classes (e.g., Cropland-LC, Agriculture-LU), where the LUCAS nomenclature consists of a detailed hierarchy of types that spans 3 levels of detail. When the more detailed classification of LC and LU is considered, i.e., levels 2 (e.g., Cereals-LC) and 3 (e.g., Maize-LC), the agreement between the crowd and LUCAS data was much lower. In the Supplementary Information, we have now added Table S5 with the complete set of LUCAS LC classes across all levels for clarity.

  1. Line 217: A near-realtime quality assurance system has been introduced, however, it is not clear, are they operated by experts or also via crowdsourcing? If by experts, how much work efforts are needed to fulfil the goal reply within 24 hours? If via crowdsourcing, how good the users can perform such a three-level LULC classification task based on images?

Response:

In the last paragraph of section 2.1 after Figure 3, we have introduced a clarification stating that a professional surveyor based at IIASA provided feedback to the campaign participants. The surveyor had a temporary full-time contract for the duration of the campaign, and the costs of this contract are included in the cost comparison that we provide in the last section of our manuscript. In terms of effort, the customized Geo-Wiki branch (now shown in the Supplementary Information) allowed the surveyor to examine the location of incoming contributions, the pictures and the answers in order to provide direct feedback to the participants.

  1. Line 292: please check whether the formula (1) is correctly presented with only one left bracket. Since a formula presented with the double colon is also not often to be seen, is there any other research also used such form to present random/fixed effect?

Response:

The formula had a mistake, with a parenthesis missing, which we have now corrected. It uses the same notation that we employed in our analysis of the FotoQuest Campaign of 2015 (Laso Bayas et al., 2016), and the random effect notation is adapted from the one described by Piepho, Büchse and Richter (2004) “A mixed modelling approach for randomized experiments with repeated measures”.

  1. From formula 1, are the categorical variables (e.g. SKIP, HOM) and continuous variables (e.g. DFQ, QpU) considered differently for the evaluation? If yes, how? If not, what is the reason they do not need to be separately considered?

Response:

We are not sure what the reviewer means by separate treatment of categorical and continuous variables. SKIP is a dummy variable, indicating that the participant skipped (i.e., did not reach the point) =1 or did not skip (i.e., reached the point) =0. HOM is a categorical ordinal variable with 4 categories, indicating the distance at which the land cover changes, with values of: <1.5, 1.5 - 10, 10 – 50, and >50 meters. These, together with the “flagged points” variable (FP – previously called high-quality points- dummy 1/0), are considered as classes or categories. The only continuous variable are distance from the quest to the actual point coordinates (DFQ in meters) and the number of points a user has sent (QpU). We have added short descriptions to each variable in the paragraph before the formula. They are considered simultaneously in the multivariate model in order to understand the effects of each of them individually, after controlling for all the others.

  1. Line 324: the reason why cropland has been paid special attention than others is not clear at the beginning, which should be explained before presenting the results.

Response:

We have added the reason in the first paragraph of the methodology as follows: “There is emphasis on the Cropland class in this paper because in the 2015 study, Cropland was the class with the most disagreement”.

  1. Line 360: what happened to the rest of the other classes? Please also provide the confusion matrix from 2015 in order to present a complete performance comparison for all classes.

Response:

Since the focus of this paper is the 2018 campaign, we chose not to show the complete 2015 matrix for comparison but instead we refer the reader to the numbers in a previous paper (i.e., Table 4, Laso Bayas et al., 2016). We have also added that FotoQuest Go performed better in all classes in the text. Additionally, for information, here we provide a comparison of the results from both campaigns for the remaining classes:

Class

Accuracy 2015

Accuracy 2018

Artificial

54

94

Cropland

60

91

Woodland

61

97

Shrubland

14

71

Grassland

57

79

Bare land

0

43

Water

50

89

 Wetlands

0

75

  1. Line 404 & 407: the results from other variables are mentioned in a less detailed way. A table/figure would be helpful to have an overview of the influence of all the variables. Further discussion of the reasons is needed to provide more insights from the authors.

Response:

The results of all significant variables (p value <0.05) are detailed in the text. The other classes mentioned were not significant, as mentioned in the text, i.e., their influence is not detectable within our data.

  1. Line 458: "Corine +" was mentioned without any further explanation, though "CORINE" was mentioned in line 49. What are the difference?

Response: Corine +, or more accurately CLC+, is the second generation of CLC and takes a very different approach to the development of the CLC, which is comprised of a backbone, a core and then instances generated from these other products. Clarification that CLC+ is the second generation of CLC was added to the text, along with a reference, so that interested users can read more about this new process.

  1. Line 274: Is Cochran-Mantel-Haenszel test the only choice to conduct a group comparison test? If not, the reason why this test was more suitable than others need to be clarified.

Response:

A traditional way to compare two groups would be through a t test or even through an ANOVA but these tests are valid for paired data. In our manuscript, we compare the agreement of one campaign (in 2015), done 3 years before, with similar but not equal conditions to a campaign in 2018, so what we are querying is whether the proportions of agreement (at different LC levels) for each campaign are significantly different. Both campaigns also have different sample sizes. For these two reasons, the CMH test was the most appropriate test for this comparison. We use the CMH to understand the likelihood of agreement of the crowd with LUCAS in one campaign compared to the other campaign, while a simple Chi square test is used to establish whether this difference is significant or not with 95% confidence (alpha=5%).

  1. Figure 6: the notions a/b/A/B/aa/bb needs to be explained in the text. It is also not clear the relationship between the mentioned ratios (2.9, 3.5, 3.1) and the numbers in this figure. More explanations are needed for the notions and numbers in this figure.

Response:

The paragraph has been slightly rewritten to further guide the reader, explaining that the comparisons between FotoQuest campaigns are done at each LC level. Additionally, the following note has been added before Figure 5: “Note that the test results and agreement ratios are valid only for comparison at each LC level but not between levels (and hence the different letter notation in Figure 5 – for LC level 1: a/b; for LC level 2: A/B; for LC level 3 aa/bb).”.

Minors

  1. Since there are many abbreviations appeared in the manuscript, a list of abbreviations (as in the template) will be very helpful for readers to track the meaning of these abbreviations.

Response: We have now added a list of abbreviations at the end of the paper.

  1. There are words with typos and inconsistent capital/lowercase letters, please have a detailed check across the whole text. E.g. Line 292 OpU/QpU, Euros/euros are not written consistently.

Response:

We have checked the manuscript for typos and inconsistent capitalization. When land cover classes from LUCAS are used, the first word in the class is capitalized throughout the paper.

  1. "Section 4.1" appeared twice at Line 385 and Line 306

Response: We have removed the second appearance of Section 4.1.

  1. Line 343: Bottom line is missing for Table 5

Response: The bottom line has now been added.

Reviewer 3 Report

Dear Authors,

Althoug I like the general topic of your research and also believe that it can be if interest for the wider audience, I see several things in your manuscript which must be adjusted from my perspective.

Regarding the structure or your manuscript I would suggest to review all sections and make sure that describing the experiment, literature review, description of the methods and conclusions for the results are not mixed between the sections.

## Introduction
Your introduction puts a strong emphasis on LUCAS and provides already a lot of information on the 2015 and 2018 FotoQuest campaigns. In my opinion a broader review of the scientific literature towards LULC on the one hand and citizen science on the other would be benefitial. I'm missing a clear scientific goal stated in the introduction. when reading the introduction, for me it was not clear which research gaps you try to tackle with your paper. Explicitly formulating one or two research questions already in this section would be needed here. Also providing a few sentences on the general structure of your paper at the end of the introduction would help readers to follow your ideas better.

## FotoQuest Go Europe
Section 2 could be streamlined in a way that you distinguish more between the app and the experiment design / campaign design. For instance it is needed that you explicitly state all the land cover and land use classes used. For me this is part of your experiment design and it should be stated there. A detailed description of the "levels" used for the land cover classification is needed as well. You refer to this at a later stage, but I have been struggling to find a place in your manuscript how these levels have been defined. (You provide examples at the beginning of your Methods section, but no definition or description.) I section 2.2. you also already provide some of the results of your campaing, e.g. "71% of the quests showed no change...". It think this could be part of your results section.

Furthermore, I do not fully understand your definition of "High Quality quests" (still belongs to 2.2?). For me this rather sounds like a research question and it would be indeed interesting to evaluate the quality of the quests in regard to some of the characteristics you describe, but not claim them having a "high quality".

In Section 2.3 I do not fully get how you want to evaluate the impact of the quality assurance system. Maybe this could have been highlighted already in the introduction.

The map is blurry. Not sure if this only due to the PDF I got or if this can be improved from the authors side.

## Materials and Methods
The "3. Materials and Methods" section is rather short. Again a more detailed description on the "levels" is needed here to understand if your methods are applicable. Your definition of "agreement" is missing. "The data from the 2018 LUCAS campaign were used to calculate the agreement with data collected from FotoQuest Go Europe at three different levels of.." There are several ways to define agreement, ideally you choose a method which will also account for the fact that labels in both data sets can be the same just by chance, e.g. by applying kappa statistics or similar methods.

As a general comment to structure your methods, I would suggest to start with the confusion matrix between LUCAS and FotoQuest 2018 and the measures based on the confusion matrix. As a next step you could take a deeper look into measures for the individual LC and LU classes. Again, here it would have been benefitial if you could provide a research question in the beginning of your manuscript. This would make it easier for the reader to assess how applicable your methods are.

Regarding the statistical model you apply, I'm not familiar with the concept of LC homegeneity. Is this something that you defined for this research or is there any reference for it? As mentioned already earlier, I would not use "high quality point" as a parameter in your model, but rather try to directly use the characteristics of the contribution you initially use to assess if a point is considered high quality.

Provide more information on the surveys you conducted and how you are going to analyse the data.

## Results
Results section should be written more in context to the research questions. At the current state many different things are described and it is not always clear how they refer to each other. I think starting with the confusion matrix could be one of the first results. After presenting this you can go into detailed description for each class and for the individual levels.

section 4.1 appears twice (line 306, line 385)

some sentences are really hard to understand, e.g. line 341: "In particular, for level 3, the accuracy of locations with change at level 3 are considerably higher than those when considering all points while there is a slight decrease when considering level 1". It would be good to got through the entire section and avoid such long and complicated sentences.

Table 4: I find it confusing that you don't use the % in the table. Maybe it would be easier to read if you provide the overall agreement as float and then put n in brackets, e.g. 0.49 (174) for the first entry

line 375: It seems that you are using a different definition for agreement when it comes to LU. If so this must be described in the methods section already. Currently, you consider that both datasets agree if any of the classes match. What would happen if you choose a stricter approach?

line 396ff: it would be good to put the results of the statistical model in a table for easier comparison of the effect of each parameter.

Section 4.3: at the current state there is no real connection between the user study and the results of the campaign. How does the user survey relate to the data output and quality of the data?

## Discussion
The discussion section should provide more references to other similar citizen science projects and should discuss the results of this study in the context of the existing literature. Most conclusion are really broad and lack further references

line 454: "potential of citizen science when mobile technology is used in a positive, well designed way." what does "positive" refers to in this context?

line 474: "where a virtual reality AI-enhanced system, combined with additional information delivered in an easy-to-understand way, could help to improve identification of this difficult LC type". I do not get, what is the meaning of "virtual reality AI-enhanced system"?. Ideally try to make shorter sentences and avoid using too many buzz-words without references.

line 491: "IIASA is 493 also currently experimenting with augmented reality games" can you provide any references for this?

I miss a critical reflection on the methods your used in your research. In the discussion section you should provide more directly the limitations of the statistical models, definitions for agreement etc.

Author Response

We would like to thank the reviewer for their very useful comments, which have helped to improve the paper. The reviewers’ comments are in bold followed by our response.

Although I like the general topic of your research and also believe that it can be if interest for the wider audience, I see several things in your manuscript which must be adjusted from my perspective.

Regarding the structure or your manuscript I would suggest to review all sections and make sure that describing the experiment, literature review, description of the methods and conclusions for the results are not mixed between the sections.

Response:

We have reviewed the whole manuscript and refocused it on the comparison of the FotoQuest results with LUCAS. We have now moved the details of the feedback system and the results of the user survey to Supplementary Information, as well as providing additional clarifications to each section.

## Introduction
Your introduction puts a strong emphasis on LUCAS and provides already a lot of information on the 2015 and 2018 FotoQuest campaigns. In my opinion a broader review of the scientific literature towards LULC on the one hand and citizen science on the other would be benefitial. I'm missing a clear scientific goal stated in the introduction. when reading the introduction, for me it was not clear which research gaps you try to tackle with your paper. Explicitly formulating one or two research questions already in this section would be needed here. Also providing a few sentences on the general structure of your paper at the end of the introduction would help readers to follow your ideas better.

Response:

Since we have now refocused our paper on comparing the results from the campaign with  LUCAS, the main research question is: can the general public produce high quality results using the FotoQuest mobile app when compared with the authoritative LUCAS survey data?  Additionally, the last paragraph of the introduction section has been modified to include a clear hypothesis and what we want to test. The removal of the two sections mentioned above make the structure of our manuscript much clearer (and simpler).

## FotoQuest Go Europe
Section 2 could be streamlined in a way that you distinguish more between the app and the experiment design / campaign design. For instance it is needed that you explicitly state all the land cover and land use classes used. For me this is part of your experiment design and it should be stated there. A detailed description of the "levels" used for the land cover classification is needed as well. You refer to this at a later stage, but I have been struggling to find a place in your manuscript how these levels have been defined. (You provide examples at the beginning of your Methods section, but no definition or description.) I section 2.2. you also already provide some of the results of your campaing, e.g. "71% of the quests showed no change...". It think this could be part of your results section.

Response:

Examples of the 3 hierarchical levels are now shown in the introduction (pre-last paragraph). The separation of the feedback system allows to have a clear distinction between the app and the campaign (experimental design). Since we now stated we will compare against LUCAS, the classes are specifically those from LUCAS. Furthermore, we have now added Table S5 to the Supplementary Information with the complete all full set of LUCAS LC classes across all levels for clarity.

Furthermore, I do not fully understand your definition of "High Quality quests" (still belongs to 2.2?). For me this rather sounds like a research question and it would be indeed interesting to evaluate the quality of the quests in regard to some of the characteristics you describe, but not claim them having a "high quality".

Response:

We have renamed the high quality points as “flagged points”, and we have described what this means in more detail at the end of section 2.

In Section 2.3 I do not fully get how you want to evaluate the impact of the quality assurance system. Maybe this could have been highlighted already in the introduction.

Response:

This has now been removed from the main text and shifted to the Supplementary Information as we are not evaluating this in the current manuscript.

The map is blurry. Not sure if this only due to the PDF I got or if this can be improved from the authors side.

Response:

The map we submitted was high resolution. The pdf version does show a lower quality figure, and due to the density of points, it is also difficult to display. We will ensure that a high-resolution figure is sent to the journal for production purposes.

## Materials and Methods
The "3. Materials and Methods" section is rather short. Again a more detailed description on the "levels" is needed here to understand if your methods are applicable. Your definition of "agreement" is missing. "The data from the 2018 LUCAS campaign were used to calculate the agreement with data collected from FotoQuest Go Europe at three different levels of.." There are several ways to define agreement, ideally you choose a method which will also account for the fact that labels in both data sets can be the same just by chance, e.g. by applying kappa statistics or similar methods.

Response:

Given the new focus of the paper, we have reorganized the methods section, stating clearly in the first paragraph how we define agreement and providing examples of the three different LC levels.

As a general comment to structure your methods, I would suggest to start with the confusion matrix between LUCAS and FotoQuest 2018 and the measures based on the confusion matrix. As a next step you could take a deeper look into measures for the individual LC and LU classes. Again, here it would have been benefitial if you could provide a research question in the beginning of your manuscript. This would make it easier for the reader to assess how applicable your methods are.

Response:

In the same first paragraph, after defining what we mean by agreement, we mention the confusion matrix and why the results are disaggregated in the way that they are, e.g., why a particular emphasis was put on cropland.

Regarding the statistical model you apply, I'm not familiar with the concept of LC homegeneity. Is this something that you defined for this research or is there any reference for it? As mentioned already earlier, I would not use "high quality point" as a parameter in your model, but rather try to directly use the characteristics of the contribution you initially use to assess if a point is considered high quality.

Response:

We used this concept in our previous FotoQuest vs LUCAS comparison (Laso Bayas et al., 2016), but in this campaign we just asked the participants how far from the quest exact location was the nearest different LC. We have now made clearer the homogeneity categories and how they are used in the model. The high-quality point variable has been renamed as flagged points and it is an additional dummy variable to control for.  

Provide more information on the surveys you conducted and how you are going to analyse the data.

Response:

We have moved the whole survey section to the Supplementary Information since this is no longer the focus of the paper. However, we are interested in understanding whether some user characteristics may correlate to how well (or poorly) they perform in the campaign, e.g., how many flagged points a certain user provided compared to others.

## Results
Results section should be written more in context to the research questions. At the current state many different things are described and it is not always clear how they refer to each other. I think starting with the confusion matrix could be one of the first results. After presenting this you can go into detailed description for each class and for the individual levels.

Response:

Since the research question is now focused on agreement with LUCAS, the sequence in which the results are presented should make more sense.

section 4.1 appears twice (line 306, line 385)

Response: We have corrected this in the text.

some sentences are really hard to understand, e.g. line 341: "In particular, for level 3, the accuracy of locations with change at level 3 are considerably higher than those when considering all points while there is a slight decrease when considering level 1". It would be good to got through the entire section and avoid such long and complicated sentences.

Response:

We have gone through the entire manuscript and shortened these types of long sentences where relevant.

Table 4: I find it confusing that you don't use the % in the table. Maybe it would be easier to read if you provide the overall agreement as float and then put n in brackets, e.g. 0.49 (174) for the first entry

Response:

We used the same format as in our previous publication (Laso Bayas et al., 2016) where we also compared FotoQuest and LUCAS for agreement, to make it easy to refer to both studies and derive conclusions.

line 375: It seems that you are using a different definition for agreement when it comes to LU. If so this must be described in the methods section already. Currently, you consider that both datasets agree if any of the classes match. What would happen if you choose a stricter approach?

Response:

Indeed, the agreement calculation is different, mostly due to the focus on LC. We have modified the methods section and added the following to the first paragraph:

“Here we define LU agreement occurring when any of the classes selected by users match those mentioned in LUCAS. We did not include all LUCAS LU classes, just the most common ones and only those from LU level 2. This was done purposefully so that respondents were not overwhelmed by the task and because we wanted to focus on LC instead.”

line 396ff: it would be good to put the results of the statistical model in a table for easier comparison of the effect of each parameter.

Response:

There were 3 models run, one for each LC level. Out of these, only the models that were run for levels 2 and 3 have a single parameter each that showed a significant effect for the agreement between LUCAS and FotoQuest, i.e., with a p value lower than 0.05. Consequently, comparing the effects between all parameters would not be valid and, therefore, we only report the corresponding change of odds of agreement caused by the selected significant parameters in each model.

Section 4.3: at the current state there is no real connection between the user study and the results of the campaign. How does the user survey relate to the data output and quality of the data?

Response:

The user study has been removed from the main text and provided as Supplementary Information.

## Discussion
The discussion section should provide more references to other similar citizen science projects and should discuss the results of this study in the context of the existing literature. Most conclusion are really broad and lack further references

Response:

We have added references to other citizen science products and framed parts of the discussion to better reflect the existing literature.

line 454: "potential of citizen science when mobile technology is used in a positive, well designed way." what does "positive" refers to in this context?

Response: This text has now been rewritten and the word positive removed.

line 474: "where a virtual reality AI-enhanced system, combined with additional information delivered in an easy-to-understand way, could help to improve identification of this difficult LC type". I do not get, what is the meaning of "virtual reality AI-enhanced system"?. Ideally try to make shorter sentences and avoid using too many buzz-words without references.

Response:

We have rephrased this sentence: “Perhaps an AI-enhanced system, i.e., a system that provides automated crop identification to the user in real-time using techniques from computer vision, could help to improve identification of this difficult LC type.”

line 491: "IIASA is 493 also currently experimenting with augmented reality games" can you provide any references for this?

Response:

We do not have any references to this ongoing work, so we have removed this from the text. However, for information, there is a small element of augmented reality in the latest FotoQuest app in which arrows are added to the picture to help users take photos in the N, S, E and W directions.

I miss a critical reflection on the methods your used in your research. In the discussion section you should provide more directly the limitations of the statistical models, definitions for agreement etc.

Response:

We have added the following paragraph before the end of the last section:

It should be noted that the study is limited to agreement as defined above, i.e., the match between LC classes, which is a standard method for measuring agreement. Methodologically speaking, the performance of FotoQuest could also have been analyzed in a different way, e.g., proximity to the target, the number of citizens visiting hard to access points or locations that were only photo-interpreted in LUCAS, etc. We are currently looking at alternative measures of system performance and potential improvements for future campaigns. Furthermore, a limitation of the employed analysis (i.e., generalized linear mixed models) is that with large sets of data, the computations take quite some time and the models may not converge, which could be a problem in the future when a larger amount of data is collected by citizens.

Round 2

Reviewer 2 Report

The manuscript has gone through a significant revision as compared to the earlier version. Major issues from my side were already taken into account except for one small issue.

3. In Piepho, Büchse and Richter (2004), colons are used in the form of “fixed : random”, which is slightly different than the “::” notion presented in the manuscript.

Author Response

The manuscript has gone through a significant revision as compared to the earlier version. Major issues from my side were already taken into account except for one small issue.

Response: We thank the reviewer for their comment.

  1. In Piepho, Büchse and Richter (2004), colons are used in the form of “fixed : random”, which is slightly different than the “::” notion presented in the manuscript.

Response:

We previously replied that the notation was adapted from Piepho, Büchse and Richter (2004), but we meant to say that it was slightly modified from them. We can also change it to “:” (single colon) but we would prefer to leave it as “::” (double colon) since we used this notation already in the comparison between LUCAS and FotoQuest Austria 2015 (Laso Bayas et al., 2016). This would make it easier for readers of the previous paper to compare it with the results in the new manuscript.

Reviewer 3 Report

Dear authors,
thanks for taking the time to revise the manuscript. I think that you have been able to streamline the content a lot and this time it was much easier to follow your thoughts. I appreciate that you put some parts of the manuscript into supplementary material to make the text shorter as well.

A few comments for further improvements:

Concerning "agreement":

  • you still do not explicitly define "agreement" and use the term in a rather broad context
  • the term "agreement" is used very often in your manuscript, however you do only say that "agreement has been calculated at different levels..." (line 217ff.)
  • from my understanding it seems that you use the term "agreement" when you refer to "accuracy", e.g. generated from a confusion matrix by deriving the number of matching classifications divided by the overall number of classifications
  • in the discussion you refer to this as "the standard method for measuring agreement" (line 463) --> when it comes to evaluating data with the help of a confusion matrix agreement is often calculated by cohens kappa or other measures of inter-rater reliability, but I think that this is not what you do in your manuscript
  • please make sure to use a more precise term than "agreement" in your manuscript and provide a clear reference for it

Results from the Foto Quest 2018 campaign:

  • add a sentence providing an overview on contribution inequality among users --> this will be important for the reader since if will very likely effect your statistical analysis
  • is contribution inequality among users of concern for your study? e.g. has most data been contributed by a single or very few users only? --> is so this should be discussed and highlighted as a limitation, if not even better

Author Response

Dear authors, thanks for taking the time to revise the manuscript. I think that you have been able to streamline the content a lot and this time it was much easier to follow your thoughts. I appreciate that you put some parts of the manuscript into supplementary material to make the text shorter as well.

Response: We thank the reviewer for their comment.

A few comments for further improvements:

Concerning "agreement":

  • you still do not explicitly define "agreement" and use the term in a rather broad context

Response:

We have now defined the terms percent agreement and modelled agreement. Percent agreement is the simple agreement between classes that could be obtained from, e.g., a confusion matrix. Modelled agreement is used in the binomial multivariate regressions and it can only take values of 0 (disagreement) or 1 (agreement). We have re-written the methods section to reflect these two definitions and checked the subsequent text for consistency of use. The percent of agreement metric has been used so that the results are comparable with our previous comparison of the LUCAS and FotoQuest systems (Laso Bayas et al., 2016).

  • the term "agreement" is used very often in your manuscript, however you do only say that "agreement has been calculated at different levels..." (line 217ff.)

Response:

We hope that the definitions of agreement (see previous answer) help clarify the text, but we have also gone through the manuscript to make sure that we refer to agreement in a consistent way.

  • from my understanding it seems that you use the term "agreement" when you refer to "accuracy", e.g. generated from a confusion matrix by deriving the number of matching classifications divided by the overall number of classifications

Response:

Since agreement is now better defined (see previous 2 answers), we hope this is clearer now. We use percent agreement as a simple measure to align with the same description in our previous manuscript (Laso Bayas et al., 2016).

  • in the discussion you refer to this as "the standard method for measuring agreement" (line 463) --> when it comes to evaluating data with the help of a confusion matrix agreement is often calculated by cohens kappa or other measures of inter-rater reliability, but I think that this is not what you do in your manuscript

Response:

We have removed the “standard method for measuring agreement” statement.

  • please make sure to use a more precise term than "agreement" in your manuscript and provide a clear reference for it

Response:

We have now defined percentage and modelled agreement (see previous responses).

Results from the Foto Quest 2018 campaign:

  • add a sentence providing an overview on contribution inequality among users --> this will be important for the reader since if will very likely effect your statistical analysis

Response:

In the methods section, just before equation 1, we added this text: “Given that different users provided dissimilar amounts of data, i.e., some users visited more locations than others, the QpU variable was introduced to control for the unequal contribution of observations per user.” The unbalanced nature of the data set was therefore controlled by the variable QpU.

  • is contribution inequality among users of concern for your study? e.g. has most data been contributed by a single or very few users only? --> is so this should be discussed and highlighted as a limitation, if not even better

Response:

Please see previous answer. This was controlled for by the variable QpU. The variable did not significantly affect the results.